# The Possible Role of Pathogens and Chronic Immune Stimulation in the Development of Diffuse Large B-Cell Lymphoma

**DOI:** 10.3390/biomedicines12030648

**Published:** 2024-03-14

**Authors:** Lajos Gergely, Miklos Udvardy, Arpad Illes

**Affiliations:** Department of Hematology, Institue of Internal Medicine, Faculty of Medicine, University of Debrecen, H-4032 Debrecen, Hungary; mudvardy@med.unideb.hu (M.U.); illes.arpad@med.unideb.hu (A.I.)

**Keywords:** DLBCL, pathogenesis, B-Cell Receptor, microenvironment, pathogens, EBV, HHV8, HCV, HBV, HIV, retrovirus, *helicobacter*, *campylobacter*

## Abstract

Diffuse large B-cell lymphoma (DLBCL) is the most common type of non-Hodgkin lymphoma. The disease is very heterogeneous, with distinct genetic alterations in subtypes. The WHO 2022 5th edition classification identifies several minor groups of large B-cell lymphoma where the pathogenetic role of viruses (like EBV and HHV-8) is identified. Still, most cases fall into the group of DLBCL not otherwise specified (NOS). No review focuses only on this specific lymphoma type in the literature. The pathogenesis of this entity is still not fully understood, but several viruses and bacteria may have a role in the development of the disease. The authors review critical pathogenetic events in the development of DLBCL (NOS) and summarize the data available on several pathogenetic viruses and bacteria that have a proven or may have a potential role in the development of this lymphoma type. The possible role of B-cell receptor signaling in the microenvironment is also discussed. The causative role of the Epstein–Barr virus (EBV), human herpesvirus-8 (HHV-8), Hepatitis C virus (HCV), human immunodeficiency virus (HIV), Hepatitis B virus (HBV), and other viruses are explored. Bacterial infections, such as *Helicobacter pylori*, *Campylobacter jejuni*, *Chlamydia psittaci*, *Borrelia burgdorferi*, and other bacteria, are also reviewed.

## 1. Introduction

Diffuse large B-cell lymphoma (DLBCL) is the most common aggressive non-Hodgkin lymphoma in the WHO large B-cell lymphoma category. There are several published reviews on pathogens and signaling mechanisms, but no concise review is found in the literature focusing on only the most common aggressive lymphoma, DLBCL. Some pathogens have a proven association with DLBCL, but others have only reported cases and postulated roles in the development of this lymphoma. The authors provide a review of the essential signaling pathways and possible pathogens involved in the development of this disease.

The histological picture of DLBCL is characteristic of large, activated B-cells, so-called centroblasts, and centrocytes in the Kiel classification. During the last 50 years, several classification systems were developed to characterize this entity, and in the latest 2022 WHO classification, more than 25 subtypes were listed [1]. Despite all these entities, most are minor groups with distinct clinical characteristics, but most cases fall within the DLBCL NOS (not otherwise specified) group. This is still a very heterogeneous group with different outcomes for individual patients. Alizadeh and colleagues described two significant subgroups based on gene expression profiling in 2000 [2]: the germinal center type (GCB) and the activated B-cell type (ABC), where the latter had a worse clinical outcome, as only 16% of patients were alive at 5 years compared to 76% in the GCB group. Utilizing these two groups could still not significantly improve the treatment results; only the addition of rituximab improved survival in both subgroups, but the difference was still significant. Several clinical trials with novel agents addressed these two subgroups with no significant improvement over the standard R-CHOP (rituximab-cyclophosphamide-adriablastine-vincristine-prednisone) treatment.

In 2018, at least five distinct subgroups were identified by next-generation sequencing in two separate publications [3,4]. The subgroup divisions mainly focused on the pathogenic mutations like MYD88, EZH2, NOTCH1, NOTCH2, BCL-6, BCL-2, CD79B, and TP53, showing different survival outcomes in each group. With all these data available, we are gathering more and more knowledge on the pathogenetic mutations beyond the malignant B-cell proliferation, and groups that were previously homogenous are subdivided into distinct subgroups with specific mutations that novel drugs can target. The technique is not widely used in routine clinical practice. This constant evolution is slowly improving the survival of DLBCL patients. The selection of available targeted treatments is not based on these identified mutations as they are not universally tested in all patients, and we need clinical trials proving the benefits of specific targeted therapies in particular mutation-bearing subgroups. Besides all these data, we still need an understanding of the initial pathogenesis and possible pathogens involved in developing this disease. This may enable us to identify this lymphoma early and treat it with more specific therapy, preventing progression into more aggressive forms that still cannot be treated with 100% efficacy.

The authors try to summarize the critical pathogenetic events in the initial development of DLBCL NOS that may have a role in pathogen-associated lymphomas. The authors did an extensive literature search to explore all published cases where infections were documented in the pathogenesis of DLBCL. Only infections with a possible role in disease development were included; no cases were included where infection occurred during therapy and may have been associated with therapy-induced immune suppression. The authors provide data on whether each found pathogen has a proven role or if its association is documented. However, the role still has to be confirmed, as only a few cases are reported, and the association may be accidental. They highlight likely pathogens in the evolution of this lymphoma type.

The survival of malignant B-cells is a complex multifactorial process but requires mutations and alteration in signaling mechanisms involved in the pathogenesis of DLBCL. The review cannot list all pathogens, but the authors try to list pathogens that have been documented to be present in this lymphoma type. Also, possible pathogens are mentioned. However, some associations are only postulated based on a few reported cases; thus, later reports may alter these findings. Pathogens with a likely association are also mentioned, but no reported cases are mentioned in the literature.

## 2. B-Cell Receptor Signaling and the Germinal Center Reaction

The B-cell receptor (BCR) consists of two immunoglobulin heavy and light chains linked to the CD79A and CD79B complexes. It is the primary functional receptor of B-cells, helping them mature into antigen-specific immune cells of the adaptive immune response. The immunoglobulin receptor alone can only induce low-level “tonic” signaling required for B cell survival, but upon engaging the immunoglobulin complex to the CD79A/B transmembrane molecules, the intracellular immunoreceptor tyrosine-based activation motif (ITAM) domains initiate a phosphorylation cascade that produces a strong signal through phosphoinositol-3 kinase and mTOR (mammalian target of rapamycin), the CARD11-BCL10-MALT1 (CBM) complex, and the IkappaB-kinase (IKK) complex, eventually leading to nuclear-factor-kappaB (NF-kappaB) signaling and proliferation [5]. The expression of the surface CD10 helps B-cells remain in the germinal center, where bcl-6 expression augments low-level BCR signaling. This is called the germinal center reaction, where the B-cells are selected based on the receptor antigen specificity. Leaving the germinal center, the expression of CD10 is lost, bcl-6 is downregulated, and only B-cells that do not react with self-antigens survive in normal conditions. This reaction is crucial in normal B-cell development, and any disturbance in the very complex process may result in pathologic B-cells leaving the germinal center and proliferating [6]. The NF-kappaB-induced upregulation of interferon regulatory factor-4 (IRF-4) transcriptional factor is crucial in downregulating the bcl-6 (B-cell lymphoma 6) expression [7]. The pathogenesis of germinal center type DLBCL mostly depends on tonogenic BCR signaling, as the cells maintain the germinal center phenotype and have molecules augmenting the low-level B-cell receptor signaling, which is not antigen-dependent. However, in the activated B-cell type, the BCR signaling alone is not enough for survival; CD79A and CD79B signaling are also required for the survival of pathologic B cells. This signaling requires antigens to be recognized by the B-cells. In the case of CD79A and B gain of function mutations, a constant signal is produced without antigenic stimulation, thus making the B cells antigen-independent [6]. Downstream mutations of the signaling cascade (MYD88, CARD11) may also maintain constant B-cell activation without antigenic signaling.

## 3. Toll Receptor Signaling

Toll receptors on the surface of B-cells are the critical elements of the innate immune system. They recognize conservative pathogen-associated molecular patterns and alternatively activate the B cells mainly through the MYD88 (myeloid differentiation primary response 88) and interleukine-1 receptor-associated kinase 4 (IRAK4), eventually interacting with the nuclear-factor kappaB kinase (IKK) complex. Elements of downstream signaling are shared with the BCR-initiated cascade signaling. The Toll-receptor signaling also initiates cytokine production and T-B cell interactions to promote the immune response. The Toll-4-like receptor recognizes the surface glycosylphosphatidylinositol (GPI) anchor of pathogens. Its overexpression in some DLBCL lines has been described [8]. Its importance is that it has the most robust intracellular signaling mechanism, producing a solid proliferating signal. The most important is the Toll-9-like receptor that recognizes conservative cytosine–phosphate–guanine (CpG) dideoxynucleotide motif sequences present in many pathogens (Gram-positive and negative bacteria, DNA viruses), making it the most important universal Toll-like receptor in innate immunity. Its signaling is also through the MYD88 complex. The TLR-3 receptor binds to double-stranded RNA, and TLR-7/8 recognizes single-stranded RNA. The schematic pathways in TLR signaling are presented in Figure 1. The constant Toll-like signaling maintains an MYD88 activation, making way for possible MYD88 mutation in the CBM complex, which is present in several DLBCL cases [9,10].

## 4. Microenvironment and the Role of Macrophages

It has been shown that increased CD68 positive macrophage infiltration in DLBCL represents a worse prognosis [11]. The resting M0 macrophages can be activated in a classical way by the Th1 cells to become M1 macrophages that promote the immune response through antigen presentation and cytokine production. The alternatively activated M2 macrophages through the Th2 cells are immunosuppressive, helping to control the immune response. IL-4 and IL-13 are the most critical cytokines for M2 macrophage activation and presence. The increased number of M0/M1 CD68-positive macrophages correlates with a worse prognosis in DLBCL [12]. The checkpoint molecules programmed death protein 1 and its ligand (PD-1, PD-L1) modulate the immune clearance of malignant B-cells. The T-cell immunoreceptor with Ig and ITIM domain (TIGIT) expression has recently been described in DLBCL, as PD1 expression is low in DLBCL. It is known that blocking the PD-1–PD-L1 axis in DLBCL has little clinical benefit, but agents blocking the TIGIT in the animal model have a more potent effect [13].

The TIGIT on T-cells binds to CD155 on dendritic cells and macrophages with high affinity. The TIGIT pathway is complementary to the PD1 / PD-L1 inhibitory pathway. Some pathogens may increase the TIGIT expression on T-cells, directly paralyzing them to eradicate the infected cells. Tumor-infiltrating T-cells have increased TIGIT expression, inhibiting their interaction with the malignant cells. The microenvironment of DLBCL is mainly maintained by the malignant B-cells paralyzing the immune-effector T-cells, making them unable to eradicate the disease effectively. This is maintained mostly through cytokines, but adenosine levels also paralyze the T-cells through their adenosine receptors. The JAK/STAT signaling of cytokine receptors that involves Janus-kinases (JAK), the signal transducer, and the activator of transcription proteins (STAT) enzymes through the IL-10 receptor family (IL-10RA and IL-10RB), the IL-6 receptor, and interferon alpha and gamma receptors modulates the interacting of B-cells with the cytokines in the microenvironment [5]. There are several different STATs (STAT1–STAT6) and three JAKs (JAK1, JAK2, JAK3). Upon activation, the phosphorylated STATs form homodimers and heterodimers and translocate to the nucleus, where they interact with other transcriptional factors and regulate the cell proliferation/apoptosis mechanisms, maintaining or suppressing the constant proliferation of the B-cells [14]. This can lead to B-cell activation or an induction of apoptosis. The signaling through intact JAK/STAT modulates B-cells through the cytokines, but mutations in the JAK/STAT complex can also initiate signaling without cytokines attaching to the receptor. Several pathogens can utilize this signaling to alter the host’s immune response.

## 5. Pathogens in the Development of Diffuse Large B-Cell Lymphoma

Table 1 lists the large B-cell lymphomas, with the reported frequency of a pathogen identified as it is associated with the disease. Table 2 lists the pathogens covered below in the review, with their proven (+) or possible (+/−) pathogenetic role of action in the development of lymphoma. A proven role is established if several cases are documented and pathogenetic mechanisms are explored. A possible role is postulated where cases have been published with postulated pathogenesis, but more data are required to confirm this mechanism. A detailed explanation of Table 2’s statements is in the text below.

## 6. Viral Infections

Viral infections can cause malignant diseases by utilizing elements of the viral genome, modulating normal cell cycle and repair mechanisms, constantly keeping the cells in an activated state, and disrupting the normal cell cycle regulation and repair mechanisms needed for genome integrity. Several viruses persist in the host in a latent form, modulating the normal cell cycle and maintaining a constantly altered immune phenotype and activation. Viruses may not only be causative agents, but their presence may also change the biology of the DLBCL.

### 6.1. Epstein–Barr Virus (EBV)

The human gamma-herpesvirus 4 (Epstein–Barr virus, EBV) is the most important in DLBCL pathogenesis. EBV positivity in DLBCL is first described in the WHO 2008 classification. In the recent WHO 2022 5th classification, there are separate entities where the presence of EBV infection is the hallmark of the disease in the EBV-positive DLBCL [1]. The EBV-positive DLBCL is separated from the DLBCL (NOS) category in all cases where the presence of EBV is detected in most of the malignant cells. This is a new category and an evolution from the EBV-positive DLBCL of the elderly [32]. EBV is a relatively large double-stranded DNA virus with a 172kb-long genome. It is protected by a protein nucleocapsid and an outer glycoprotein envelope. During active infection, the viral gp350/220 envelope glycoprotein binds to CD21 on B-cells, and gp42 promotes viral entry into the cells by forming a complex with MHC II [33]. After active infection during the latent phase, the viral genome may persist in the host cells, primarily in episomes, and viral glycoproteins may be expressed in infected cells [34,35,36]. Reactivation may occur later, primarily due to reduced immune function. About 95% of the population is infected with the virus, usually in childhood or young adulthood. The persistence of EBV may result in latent ongoing EBV infection, which is clinically undetectable but causes a variety of diseases. The Type I latent EBV infection is characterized by the expression of EBNA1 antigen only, and these cells contain a large, primarily functional virus, mainly causing endemic Burkitt lymphoma. Type II infections mostly express LMP1 and EBNA1, which are the default persistence programs. Latency Type III infections express more viral proteins that promote a proliferation signal (EBNA1-2-3A-3B-3C, EBNA-LP, and LMP1-2A-2B) [37,64]. The latency Type 0 infection is only detectable by EBER; no other antigens are present [33].

Viral persistence regulates B-cell proliferation through direct genetic modifications and epigenetic reprogramming mechanisms, eventually leading to several malignant diseases.

Type III infections are the most common cause of DLBCL pathogenesis by several distinct mechanisms. It has been shown that the EBNA2 expression (present in Type III infections only) increases the PD-L1 expression of cells by miR-34a upregulation [38]. This increased expression of PD-L1 makes the cells immune to T-cell-mediated clearance, promoting tumor survival. The expression of miR-155 is also increased in infected cells, leading to mTOR activation and significantly augmenting the Toll-receptor (TLR) signaling [39]. Upon immune recognition of infected cells, they escape by undergoing a germinal center reaction, temporarily reverting to a latency Type II phenotype.

The pathogenesis of EBV-associated malignancies is very complex. However, the critical factor in this process is the EBNA glycoproteins. EBNA1 is resistant to proteasomal degradation and can inhibit the p53/TP53 complex and MDM2, increasing cell survival. EBNA2 is a nuclear transcriptional factor essential for the viral program to work, but it also upregulates c-myc and CD23. Together with EBNA-LP, it can replace the intracellular part of Notch, repressing the regulatory signaling and escaping from the innate immune system [33]. Common to all DNA tumor viruses, they can interfere with the cell’s DNA damage-repair mechanism (DDR), promoting the disruption of the host cells’ integrity and making possible oncogenic mutations [65].

Recently, a novel lymphoma type was reported, the breast implant-associated EBV-positive DLBCL; it has not been determined whether this is a separate entity or can be categorized into the EBV-positive DLBCL group [66].

The complexity of latent EBV infection and its role in the malignant transformation of B cells is far more complex than described, but these are the primary mechanisms. Besides the two described entities in the WHO 2022 classification, the EBV-positive DLBCL NOS (not otherwise specified) and the EBV-positive mucocutaneous ulcer, most DLBCL patients are seropositive for EBV, indicating a previous primary infection. Based on the geographic region, the frequency of EBV-positive NOS within the DLBCL cases ranges from 2% to 15%. The association is tested with EBV-associated RNA1 (EBER) FISH testing on histological samples [67,68]. It has been documented that EBV positivity confers a significantly worse prognosis, so clinicians should be alerted. Where possible, a more aggressive therapeutic approach can be considered [69]. The endemic type of Burkitt-lymphoma is proven to be associated with EBV, but different gene expression patterns are present and have different prognoses [64]. Recently, fibrin-associated DLBCL has been described as associated with EBV infection [70]. Pyothorax-associated DLBCL belongs to the group of chronic inflammation-associated DLBCL cases. Its association with EBV is documented in some instances [71].

Further studies are required to clarify the exact role of EBV in the latter two entities. The pathogenetic role of the virus is documented in several entities listed above. Still, due to the complexity of EBV-associated altered immune cells, a more detailed analysis of these cases is needed to explore this virus further. However, it can be stated that based on the published results cited above, the presence of EBV infection at diagnosis indicates a worse prognosis for the patient compared to the EBV-negative cases. It has to be emphasized that the infection is detected on the histological sample using EBER FISH and immuno-staining of EBNA1, 2, and LMP1-2.

The prevalence of detectable EBV in DLBCL (NOS) cases varies. In a large meta-analysis, 11% of gastric DLBCL cases were associated with EBV [17]. However, among all DLBCL (NOS) cases, 4–5% were detected to be positive for the presence of EBV [15,16].

### 6.2. Human Herpesvirus 8 (HHV8)

Human herpesvirus 8 is a large double-stranded DNA virus with a genome length of 165 Kb. It is the primary cause of Kaposi sarcoma, often called the Kaposi sarcoma herpesvirus (KSHV). In the WHO 2002 classification, there are three separate entities for HHV8-associated DLBCL: the primary effusion lymphoma, the KSHV/HHV8 positive diffuse large B-cell lymphoma, and the KSHV/HHV8 germinotropic lymphoproliferative disorder [1,18,72]. HHV8-positive DLBCL (NOS) is mainly seen after the multicentric Castleman disease [72]. The pathogenesis of viral infection is complex, but one of the critical regulators is the latency-associated nuclear antigen (LANA). It is required for viral replication in the S phase, and it is a transcriptional regulator that modulates several ubiquitin ligases. It is also required for viral genome-containing episomes to attach to chromatin. This antigen is the critical modulator of viral persistence in cells, which is typical for this infection [73]. LANA directly interacts with p53 and Rb, disrupting genomic repair mechanisms [65]. Another viral protein, vIRF1, negatively regulates the ATM-dependent activation of p53 [65]. The critical element of latent HHV8 infection is through the DNA damage-response (DDR) modulation. This negative regulation alters the cell’s ability to repair detected DNA damage, which is required for viral persistence. However, disrupting these mechanisms makes the B-cells more susceptible to acquiring additional mutations and survival, which is a critical first step of malignant clone evolution [65]. The KSHV/HHV8 positive DLBCL is a rare entity that accounts for 0.1% of DLBCL cases [19]. In a recent publication, the detailed characteristics of the disease were reported, where the presence of HHV8 was detected with immunofluorescent staining on histological samples in 77 patients. The survival rate with immuno-chemotherapy first-line treatment is similar to conventional DLBCL patients. The 5-year overall survival rate was reported to be 56.3%. However, patients under 60 had significantly better survival than older patients [42].

### 6.3. Hepatitis C Virus (HCV)

The Hepatitis C virus is a small single-stranded RNA virus from the Flaviviridae family. It is primarily hepatotrophic, causing hepatic diseases like cirrhosis and cancer. However, its association with B-cell malignancy is also known in specific marginal zone B-cell lymphoma cases. The virus has two glycoproteins, E1 and E2, that play a crucial role in viral persistence by inhibiting its detection by the immune system. E2 binds to surface CD81 and modulates viral entry into the infected cells. It has been documented that HCV attaches to low-density lipoproteins (LDL) and utilizes the LDL receptor for cellular entry, escaping the immune recognition [40]. It has been shown that HCV can infect B-cells, altering their immunogenicity and making them monoclonal to produce a particular rheumatoid factor in Type II cryoglobulinemia. The early detection of these circulating B-cells in Hepatitis C-infected patients may predict later lymphoma development [41,43]. These monoclonal B-cells show an overexpression of bcl-2, which can be effectively modulated by interferon–alpha treatment [74]. The expression of the Hepatitis C virus non-structural 3 (NS3) protein is found positive in 46% of HCV-associated B-cell non-Hodgkin lymphomas, where most cases were DLBCL, comprising 36% of cases investigated [21]). The effective treatment of Hepatitis C completely reverses the B-cell clonality. Lymphoma can be prevented if treatment is started in early infection. The outcome of HCV-associated DLBCL is similar to the HCV-negative cases using anti-CD20 immunotherapy with the first-line polychemotherapy [75,76]. However, the HCV viral load at the diagnosis has a negative prognostic impact on survival [77]. The widespread use of direct-acting antiviral (DAA) drugs that effectively treat HCV infection is effective in indolent lymphomas caused by HCV. However, DLBCL cases acquired additional mutations and require the standard immuno-chemotherapy treatment, but the direct-acting antiviral therapy is safe and recommended in all patients [78]. DAA drugs effectively and safely reduce the HCV load, eradicate the underlying disease, and improve the prognosis of DLBCL patients [79]. The frequency of HCV infection in B-cell NHL varies from 6.4% to 20%, with an average of 13% calculated from a large meta-analysis [20].

### 6.4. Hepatitis B Virus (HBV)

The Hepatitis B virus is a circular, partially double-stranded small DNA virus belonging to the hepadnavirus family. The viral DNA polymerase enzyme has a reverse transcriptase activity like the retroviruses [44]. It is a hepatotropic virus, but recently, it has been documented that HBV can directly infect B-cells. The viral DNA is integrated into the lymphocyte genome, where cellular DNA polymerase enzymes make a full double-stranded DNA copy from the viral semi-double-stranded genome [36]. This integration causes the cis activation of specific genes and a distinct pattern of genetic alterations. This integration is possible due to the reverse transcriptase activity of viral DNA polymerase. The frequency of DLBCL in HBV-infected individuals is increased, and these cases occur primarily in younger adults [80,81]. It has been reported that not only chronic Hepatitis B-infected patients have an increased likelihood of developing DLBCL, but individuals who underwent HBV infection and fully recovered have an increased risk of B-cell lymphoma development as well. This is a unique finding [82]. Several authors report the possible association between Hepatitis B infection and DLBCL, and a large meta-analysis summarizing all published data reported an odds ratio of 2.06 (1.48–2.88) of DLBCL in Hepatitis B-infected individuals [83] A case of DLBCL is reported where HBV reactivation possibly triggered the onset of DLBCL [84]. The prognosis is worse than that of HBV-negative cases, highlighting the importance of viral persistence in constantly keeping the B-cell proliferative [85]. The inferior survival of HBV-infected patients with DLBCL is documented due to different biology and impaired hepatic function [45]. It has been reported that ongoing HBV infection can potentiate the resistance to S-phase arrest, causing a resistance to chemotherapeutic agents [86]. The frequency of past HBV infections in a large DLBCL cohort is reported to be 16.8% by an Italian study [22]. It has been documented that past HBV infection increases the chances of developing DLBCL with an odds ratio of 2.25 [82].

### 6.5. Human Immunodeficiency Virus (HIV)

Human immunodeficiency virus is a single-positive-strand retrovirus belonging to the lentivirus family. The virus infects the B-cells, macrophages, and microglia cells. The infection causes a characteristic immune deficiency, acquired immune deficiency syndrome (AIDS), leading to several complications. The virus can cause several B-cell lymphomas; most of the cases are DLBCL. A characteristic predilection site is the central nervous system (CNS), but several cases of systemic lymphoma have also been described. Pathogenesis is a result of the viral modulation of Bcl-2 expression, making cells resistant to apoptosis and susceptible to additional mutations. Also, the DNA damage-response (DDR) mechanism is directly altered, promotes the mutations, and also makes cells more resistant to conventional chemotherapeutic agents, resulting in worse survival of HIV-positive DLBCL cases [46,48]. Infected T-cells also present high levels of TIGIT and PD-1, making them unable to function normally and making B-cell malignancies possible [47]. The infection increases the c-myc expression level, possibly contributing to the lymphomagenesis [87]. Since the widespread use of highly active antiretroviral therapy (HAART), there has been a decline in HIV-associated DLBCL. However, HIV infection confers a worse prognosis for DLBCL patients due to several factors. These lymphomas usually bear the germinal-center phenotype with myc alterations.

Also, patients may have a decreased CD4 count, making them more susceptible to infections during the lymphoma treatment [87]. In a sizeable Chinese cohort of HIV-positive DLBCL patients, it has been reported that by administering the standard immuno-chemotherapy to these patients with the combination antiretroviral therapy (cART), the results are comparable to the HIV-negative patients’ data [46]. The 2-year PFS in HIV patients can be similar to the HIV-negative group with 75% by using cART and immuno-chemotherapy [88]. In contrast, in a recent Spanish cohort of patients from the cART era, HIV-positive DLBCL patients had a worse prognosis with decreased overall survival at 5 years of 56% compared to 74% in HIV-negative patients [89]. A recent Australian study of HIV-positive lymphoma patients reported an excellent 2-year PFS of 77% and an OS of 81% using curative immuno-chemotherapy together with cART [90]. The frequency of HIV infection in a large cohort of DLBCL patients is 5.4% in US patients [23]. The HIV infection is associated with an advanced stage in DLBCL patients, and a worse overall survival was detected in this large retrospective database study as well [23]. Another large database study reported on 115,643 NHL patients from the US; 5.9% of patients were HIV positive, but this percentage was higher in DLBCL: 7.8% [24] A SEER database study investigated HIV-associated death in non-Hodgkin lymphomas. Among all NHL-related deaths, there was a 4.6% rate in HIV-infected patients, but this was higher in DLBCL (7.3%) and in CNS lymphoma (17.6%) [91].

### 6.6. Coronavirus (SARS-CoV-2)

The human SARS-CoV-2 virus is a single-stranded RNA virus. During the recent coronavirus disease pandemic, a significant portion of the population was infected with this virus. The association of the SARS-CoV-2 infection and lymphoma is investigated, but most papers focus on COVID-19 infection and severity in lymphoma patients. There are few reports focusing on the possible pathogenic role of the virus. It has been shown that SARS-CoV-2 infection with severe respiratory symptoms is predisposed to B-cell lymphoma and DLBCL [25]. One potential mechanism is the virus-induced upregulation of miR-155, which alters the cell’s activation [49]. The virus-induced ongoing inflammation and cytokines may facilitate B-cell proliferation in these patients [50]. The infected T-cells increase TIGIT and PD-1 expression, possibly leading to a more severe form of COVID-19 [92]. This negatively impacts the immune system, making way for possible B-cell malignancies. The constant activation of the JAK/STAT-signaling pathway has been documented in patients with the SARS-CoV-2 infection, which may also contribute to malignant B-cell evolution [14]. These pathogenetic events will likely cause lymphoma, but the reported data cannot fully support this hypothesis. Besides the one study that postulates that SARS-CoV-2 infection-induced inflammation is associated with DLBCL, the virus has a clear impact on the survival of patients. Several publications highlight the importance of co-infection with SARS-CoV-2 during the treatment of lymphoma to negatively impact patients’ survival [93]. During and after the anti-CD20 treatment of these lymphomas, the impaired B-cell function negatively affects the efficacy of vaccination against SARS-CoV-2 [94].

### 6.7. Other Viruses

Human herpesvirus 6 (HHV6) and several polyomaviruses (JCPyV, BKPyV) have a role in certain diseases and malignant transformations. However, their direct role in B-cell malignancies has not been documented [51,95,96]. It has been shown that increased JCPyV and BKPyV frequency is found in the gut of DLBCL patients [97]. HHV6 is not directly linked to lymphoma, but its prevalence increases in B-cell lymphomas. Whether this is due to immunosuppression or a direct interaction of HHV6 with the B-cells is still a question. Due to the complexity of these infections, and since they all persist in human cells, expressing the surviving set of molecules, they also have a theoretical role in developing malignancy in the immune system.

GB virus-C is a single-positive-stranded RNA virus belonging to the flaviviridae group. Its association with B-cell NHL is questionable. Some reports indicate an increased incidence of lymphoma and DLBCL in infected individuals, but others could not confirm it [26,52,98].

An exciting field is the endogenous retroviruses, which may persist in the human genome during evolution. They contain transposable elements (TE), and their genome is incorporated into the human chromosomes and vertically transmitted. However, the genome is usually silenced and can only be activated by additional signals. In several documented cases, an integrated element of human endogenous retrovirus element (HREV) is detected and may be functional, making B-cells more resistant to apoptosis. The reactivation of these viruses can occur under certain conditions, and activated genomes may contain viral genes directly responsible for the malignant transformation of cells [53,54].

## 7. Bacterial Infections

Bacterial infections may cause malignancy by maintaining constant immune activation, keeping the immune system in a proliferating state and making way for possible second-step pathological mutations over time. They may also modulate the host’s immune cells, maintaining constant activation. These events require a chronic ongoing bacterial infection that may be asymptomatic; only the provoked lymphoma causes the symptoms. The chronic asymptomatic persistence of these bacteria requires that the bacterial infection alters the functional immune system. These associations are weak except for the H.pylori infection. Usually, case series are reported, and pathogenesis is only postulated based on chronic inflammation and immune activation. Thus, further studies are needed to clarify the association of these infections with lymphomas, especially DLBCL.

### 7.1. Helicobacter pylori

*Helicobacter pylori* is a Gram-negative flagellated bacteria that causes the typical infection of gastric mucosa, which causes gastric ulcers. Some cases evolve into the typical gastric mucosa-associated B-cell non-Hodgkin lymphoma (MALT). The interaction between helicobacter and B-cells is indirect; chronic inflammation leads to the lymphoma-genesis. This is proven by the fact that early stages of lymphoma can be effectively cured by eradicating the helicobacter infection with antibiotics. In about 1% of *Helicobacter pylori*-infected individuals, low-grade gastric MALT lymphoma develops, but in some cases, it progresses into gastric diffuse large B-cell lymphoma [55]. The CagA bacterial protein that induces IL-8 secretion via Nf-kappaB activation is the most critical factor in infections. It also increases Bcl-2 and Bcl-XL through p38 mitogen-activated protein kinases (MAPK) and endoplasmic reticulum kinase (ERK) [55]. This process leads to a decreased apoptosis of B-cells with activation and proliferation. The other factor, VacA toxin from helicobacter pylori, is also essential, as it induces the cytolysis of cells but also activates the JAK–STAT signaling [55]. The AKT-signaling pathway is a crucial element in developing gastric DLBCL. The CagA protein in the B-cell induces genetic abnormalities, a loss of PTEN signaling, and cyclin A2 overexpression. These B-cells have nuclear Nf-kappaB, Bcl-10, and CagA overexpression [57]. Once gastric DLBCL has lost the helicobacter pylori dependence, B-cells acquire additional mutations, and these patients have a worse prognosis. The most common mutation is EZH2, which confers a worse prognosis in the mostly germinal-center Type DLBCL [99]. Other reports confirmed that once BCR signaling is affected, *H. plyori* eradication alone cannot cure the disease [56]. The association of gastric MALT/DLBCL is high with *H. pylori;* as much as 75% of cases are positive compared to de novo gastric DLBCL, where only 36% positivity was found [27]. It has been reported that de novo *H. pylori*-associated gastric DLBCL has better survival than transformed DLBCL from MALT lymphoma [100].

### 7.2. Campylobacter jejuni

Campylobacter jejuni is a Gram-negative bacteria. The infection is broad, ranging from asymptomatic carriers to acute gastroenteritis symptoms. The chronic, asymptomatic infection causes the characteristic immunoproliferative small intestine disease (IPSID), sometimes called the heavy-chain disease [58]. The infection causes plasmocytic infiltration in the mucosa, and several cases of DLBCL have been described [28,55,101]. The course of this lymphoma is very indolent and often challenging to treat. The pathogenesis is not fully understood, but the bacteria’s CDT toxin causes cell vacuolization, enabling B-cells to invade the whole mucosa. The toxin also causes DNA damage in cells, exposing these fragments to the immune system, primarily through innate Toll receptors [28,59,102]. Treating the infection with antibiotics (macrolides, tetracycline) can stop lymphoid proliferation in the small intestine [103]. Only a few cases are reported in this entity’s literature [58]. Thus, no precise data are present on the frequency of DLBCL in this infection and the disease outcome. The association is based on possible associations based on the immune dysregulation caused by the infection and a few reported cases. Besides eradication, standard immune chemotherapy can be used, but further study is needed to clarify the role of this pathogen in DLBCL and characterize the outcome of the possibly associated disease.

### 7.3. Borrelia burgdorferi

*Borrelia burgdorferi* is an obligate anaerob spirochete bacterium. The characteristic Lyme disease caused by this bacterium is zoonotic, transmitted to humans with a tick bite. If not treated, the infection progresses into a chronic form that may cause typical cutaneous lymphoid proliferation and lymphomas. Several DLBCL cases have also been described [29,104]. However, a study from Italy could not confirm any *Borrelia* in cutaneous B-cell lymphoma patients [105]. Thus, the association is questionable, and the pathogenesis is unclear but relies primarily on chronic inflammation. The treatment of *Borrelia* with antibiotics eradicates the early-stage lymphomas, just as in Helicobacter-associated gastric lymphoma. During the disease, atypical lymphoid follicles form in the skin. The B-cells acquire Bcl-2 upregulation, making them constantly active, which may lead to additional mutations and lymphoma [29,55]. There are no clinical data on the prognosis of this disease, as only a few cases are reported. Most cases are MALT lymphoma, and only a few DLBCL-like cases have been identified. As for all infection-associated MALT lymphomas, antibiotic treatment with lymphoma-directed immuno-chemotherapy is recommended [106].

### 7.4. Chlamydia psittaci

*Chlamydia psittaci* is an intracellular bacteria acquired mainly through inhalation. It is associated with the ocular adnexal B-cell MALT lymphoma, but some cases may evolve into DLBCL [107]. However, no association of DLBCL cases was found in this reported cohort with chlamydia infection [107]. A recent Japanese study also found no correlation in a cohort of 150 patients [108]. Thus, the association of Chlamydia psittaci infection with DLBCL is still questionable. The possible hypothetical pathogenesis is based on the fact that marginal zone lymphomas may eventually evolve into DLBCL as chronic antigenic stimulation helps B-cells acquire additional mutations [60]. One study extensively investigated the mutational profile of these cases and found DLBCL-specific mutations, which affected the NFkappa-B pathway in most cases. Still, these cases were negative for chlamydia infection [107]. In a study, 36% of lymphoma cases were positive for the MYD88 mutation [61]. The detected mutations may promote the progression of the marginal zone lymphoma into DLBCL, but this still has to be confirmed. Based on the available literature, the role of this pathogen is only hypothesized and has to be further analyzed in additional clinical trials [109]. The clinical course of cases needs to be clarified; there needs to be consistent data on cases so that concise information can be provided on the prognosis and pathogenesis.

### 7.5. Other Bacteria

*Coxiella burnetti* is a Gram-negative obligate intracellular bacterium. It usually spreads with inhalation and causes the typical Q-fever. However, recently, it has been reported that chronic infection may lead to B-cell lymphomas. Cases of DLBCL have also been described [110]. In an extensive database of Q-fever patients, 0.48% developed DLBCL [30]. No data are available on the frequency of *Coxiella burnetti* in DLBCL patients. The pathomechanism is unclear, but it is postulated that the disease maintains a polarized Th1-type immune response with IFN-gamma downregulation. The overt production of proinflammatory IL-10 and IL-6 may maintain a constant B-cell activation, causing lymphomas [111].

Recently, it has been demonstrated that chronic *Escherichia coli* infection in the bladder may cause MALT lymphoma [62]. No cases of DLBCL have been described so far. Still, due to the constant inflammation, bladder DLBCL is likely to be identified in some instances, as it has been shown that MALT lymphomas can eventually transform into DLBCL [112,113]. It has to be emphasized that *E. coli* is a widespread bacterium that is often found in the urine, so further studies are needed to clarify its role in this lymphoma type.

*Achromobacter xylosoxidans* is a Gram-negative bacterium. It may be associated with pulmonary MALT lymphoma, as reported in one cohort, with some cases developing into DLBCL [63]. However, this has to be further analyzed, as other reports found only a few cases in Japanese patients. Only two cases of DLBCL are confirmed in this cohort [31]. The pathomechanism is unknown; it is possibly a chronic infection with constant immune stimulation leading to B-cell proliferation and lymphoma.

## 8. Conclusions

The pathogenesis of DLBCL is a very complex process that still needs to be fully understood. Infections may play a role in some cases. The authors provide a comprehensive review of all possible and proven pathogens involved in this disease’s etiology and highlight the essential pathways for the interaction of the B-cells with these infectious agents. As described in this review, several viral and bacterial pathogenic agents have already been identified as having a possible role in lymphomagenesis by understanding the genetics of distinct subtypes of DLBCL. Utilizing next-generation sequencing (NGS) in tumor tissue and liquid biopsy samples, the mutations involved in DLBCL are better characterized, defining very different prognostic groups with distinct pathological processes beyond. This can be correlated with the possible pathogenetic events in viral and bacterial infections and their effect on the tumor cells and the immune system. These techniques may help us explore whether more pathogenetic agents can initiate this heterogeneous lymphoma type. However, this type of data associated with pathogens is not available in the literature; future studies are needed to explore this field. Another interesting question is, due to the several subtypes of DLBCL, is there a role of any pathogen in one DLBCL subtype converting to another? It has been proven that these pathogens initially may cause indolent lymphomas but later progress into aggressive lymphoma, mostly DLBCL. However, no data are reported in the literature on pathogens involved in the dynamics of DLBCL, converting between different subtypes of the disease. This postulated event is possibly due to the direct interaction of the pathogen with the B-cells, and the maintained immune activation leads to additional mutations, but further studies are required to confirm this hypothesis. The listed viral infections, especially EBV, HIV, KSHV/HHV8, HCV, and HBV, are proven to be involved in the pathogenesis of DLBCL by not only inflammatory processes but also viral components directly interacting with the host immune cells altering their function. The association of the SARS-CoV-2 infection is confirmed in one retrospective study. Due to the fact that millions of people were infected with SARS-CoV-2, the association with DLBCL would be observable by increasing incidence worldwide if causality is present. This has to be explored in future studies.

The association of bacterial infection with DLBCL is mainly based on chronic inflammation and chronic immune stimulation by the bacterial components. Whether chronic inflammation alone may facilitate lymphoma or whether a direct interaction is needed between bacteria and the B-cells is still a question. Immune activation and BCR signaling may lead to lymphoma. Further studies are required to understand these pathogens’ exact role better and how they may cause lymphoma. Our group documented that lymphoma incidence is higher in autoimmune diseases [114], highlighting the importance of chronic immune stimulation in lymphomagenesis. This finding is further supported by detecting increased DLBCL frequency in B-cell-mediated autoimmune conditions, like Sjögren’s syndrome, systemic lupus erythematosus, and rheumatoid arthritis.

Interestingly, this association was only with DLBCL and marginal-zone lymphoma, and follicular lymphoma’s association could not be detected [115]. One interesting study demonstrated an altered mutational profile in B-cells of Sjögren’s syndrome patients, leading to DLBCL 2 years later in a case [116]. A case of DLBCL has been reported based on chronic inflammation and lobar atelectasis of the lung [117]. On the other side, it has been documented that in B-cell lymphoma patients, the antinuclear antibody titer is elevated [118,119]. Recently, it has been reported that the interferon regulatory factor 8 (IRF-8) is involved in the initial pathogenesis of DLBCL [120]). An exciting finding is that microscopic DLBCL was found in the wall of pseudocysts, highlighting the association of chronic infection with lymphomagenesis [121]. These findings highly support the idea that constant B-cell activation may eventually lead to perturbances of immune regulation and B-cell lymphoma. This mechanism may be the pathogenic process of DLBCL evolution in the listed bacterial infections. Whether this continuous activation is through a recognized self or foreign target antigen or due to alteration in the cell’s regulatory mechanisms is still a question. Still, possibly both or either mechanism is sufficient for lymphoma evolution. The authors included *E. coli* infection, which was documented to cause urinary MALT lymphoma, but no DLBCL was reported. However, transforming marginal zone lymphomas to DLBCL is a documented event [113,122]. Thus, there is a likelihood that this bacterium can cause DLBCL.

Some pathogens contribute to the pathogenesis of DLBCL and alter the prognosis of the disease. It has been documented that HIV, EBV, and HBV infection negatively impact patients’ prognoses. In contrast, HCV infection has no negative prognostic impact since using direct-acting antiviral agents [48,67,75,85]. Infection with SARS-CoV-2 does not directly alter the prognosis, but due to complications of the infection, the prognosis is worse [93].

The review highlighted the complexity of DLBCL pathogenesis and possible and proven pathogens involved in this process. Some of these associations may be incidental, and further investigation is needed. More data is required to clarify the role and pathogenetic events on SARS-CoV-2 infection in the context of DLBCL development. A systemic study is necessary to investigate the role of ongoing chronic bacterial infection and the possible development of B-cell non-Hodgkin’s lymphoma.

## Figures and Tables

**Figure 1 biomedicines-12-00648-f001:**
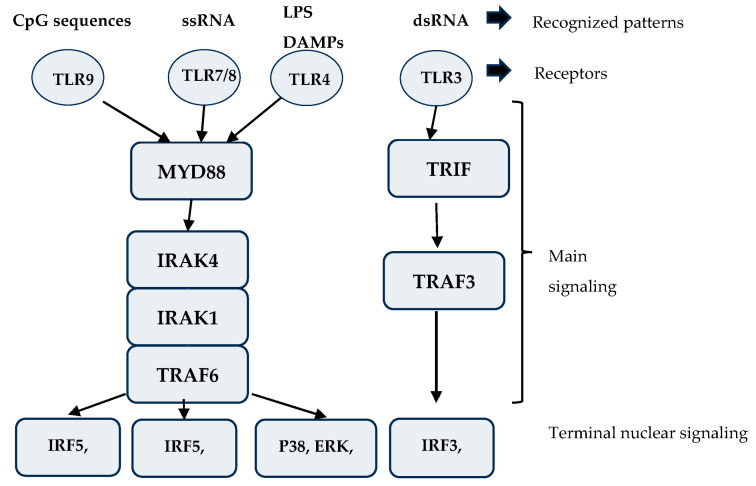
Schematic representation of Toll-like-receptor (TLR) pathways. The basic elements of the mentioned TLR signaling pathways are shown in the figure. The top row lists the recognized damage-associated molecular patterns (DAMPs). Below is the corresponding TLR. The main signaling molecules are listed as the most important terminal complex. CpG—cytosine-phosphate-guanine dideoxynucleotide motif, ssRNA—single-stranded ribonucleic acid, LPS—lipopolyscaccharide, dsRNA—double-stranded ribonucleic acid, MYD88—myeloid differentiation primary response 88, TRIF—TIR domain-containing adaptor protein, IRAK—interleukin-1 receptor-associated kinase, TRAF—tumor necorsis factor-associated factor, IRF—interferon regulatory factor, p38—p38-mitogen-activated protein kinase, ERK—extracellular signal-regulated kinase, NfKappa-B—nuclear factor kappaB.

**Table 1 biomedicines-12-00648-t001:** Association of Pathogens with DLBCL.

Pathogen	WHO 2022 Category, Lymphoma Type	Frequency (Cases Reported)	References
EBV	EBV positive DLBCL DLBCL (NOS) DLBCL (NOS)—gastric lymphoma EBV positive mucocutaneous ulcer	80–100% 4–5% 11% 100%	[1] [15,16] [17] [1]
KSHV/HHV8	Primary effusion lymphoma KSHV/HHV8-positive DLBCL KSHV/HHV8-positive germinotrophic lymphoproliferative ulcer DLBCL (NOS)	100% 100% 100% 0.1%	[1,18] [1] [1] [19]
HCV	B-cell NHL (non MALT) DLBCL	13% 36% of HCV+ lymphomas	[20] [21]
HBV	DLBCL (NOS)	16.8% (serological test)	[22]
HIV	DLBCL (NOS)	5.4% 7.8%	[23] [24]
SARS-CoV-2	DLBCL (NOS)	Severe COVID: OR:1.765 (no percentage reported)	[25]
GBV-C	NHL (all types)	4.5%	[26]
*Helicobacter pylori*	DLBCL (NOS) gastric MALT/DLBCL gastric	36% 75%	[27] [27]
*Campylobacter jejuni*	DLBCL (NOS)	1 case reported	[28]
*Borrelia burgdorferi*	DLBCL-like proliferation in the skin	5 cases reported	[29]
*Coxiella burnetti*	DLBCL (NOS)	Among Q-fever cases: 0.48%	[30]
*Achomobacter xylosoxidans*	DLBCL (NOS)	2 cases reported	[31]

The association of pathogens with diffuse large B-cell lymphoma (DLBCL). The table lists all reported associations with references and frequency or number of cases. EBV—Epstein–Barr virus, KSHV/HHV8—Kaposi sarcoma herpesvirus/human herpesvirus 8, HCV—Hepatitis C virus, HBV—Hepatitis B virus, HIV—human immunodeficiency virus, SARS-CoV-2—coronavirus 2 causing the severe respiratory syndroma of 2019/severe acute respiratory syndrome of 2019, GBV-C—GB virus C, NOS—not otherwise specified, MALT—mucosa-associated lymphoid tissue, NHL—Non-Hodgkin lymphoma.

**Table 2 biomedicines-12-00648-t002:** Pathogens and their possible pathogenetic role in diffuse large B-cell lymphoma development.

Pathogen	Inflammation	BCR Signaling	Microenvironment Modulation	Interaction with Cytoplasmic Signaling	Genetic Alteration	Genome Integration
EBV	**+** [32,33,34,35]	**+/−** [36]	**+** [37,38]	**+** [32,38]	**+** [34,36]	**+** [33]
HHV8	**-**	**-**	**-**	**+** [39]	**+** [39]	**+/−** [18]
HCV	**+** [40]	**+/−** [41]	**+** [40,41,42]	**+/−** [43]	**-**	**-**
HBV	**-**	**-**	**+/−** [20,44]	**+/−** [45]	**+/−** [20]	**+** [20,35]
HIV	**-**	**-**	**+** [46]	**+** [47]	**+/−** [22,48]	**-**
SARS-CoV-2	**+** [49]	**-**	**+** [25,50]	**+**[25]	**-**	**-**
JCPyV, BKPyV	**-**	**-**	**-**	**-**	**+/−** [51]	**+/−** [51]
HHV6	**-**	**-**	**-**	**-**	**+/−** [51]	**+/−** [51]
GBV-C	**+/−** [52]	**-**	**-**	**-**	**-**	**-**
endogenous retroviruses	**-**	**-**	**-**	**-**	**+/−** [53,54]	**+** [53,54]
*Helicobacter pylori*	**+** [55]	**+** [56]	**+** [55]	**+** [55,57]	**-**	**-**
*Campylobacter jejuni*	**+** [28,58,59]	**-**	**+/−** [59]	**+/−** [59]	**-**	**-**
*Borrelia burgdorferi*	**+** [29]	**-**	**+/−** [29]	**-**	**-**	**-**
*Chlamydia psittaci*	**+** [60]	**+/−** [61]	**-**	**+/−** [61]	**-**	**-**
*Coxiella burnetti*	**+/−** [30]	**-**	**-**	**-**	**-**	**-**
*Escherischia coli*	**+/−** [62]	**-**	**-**	**-**	**-**	**-**
*Achromobacter xylosoxidans*	**+/−** [63]	**-**	**-**	**-**	**-**	**-**

The table lists possible pathogens involved in the development of diffuse large B-cell lymphoma. The association is marked with “+” where it is likely, and there is enough literature data to support the postulation. The corresponding reference number is provided in parentheses. The mark “-” indicates that the association is unlikely and not proven. “+/−” indicates a possible association, but no sufficient data are available to confirm it, and later research may confirm the association. The references for the associations are provided in the table. BCR—B cell receptor, EBV—Epstein–Barr virus, HHV8—human herpesvirus 8, HCV—Hepatitis C virus, HBV—Hepatitis B virus, HIV—human immunodeficiency virus, SARS-CoV-2—Coronavirus 2, causing severe respiratory syndrome of 2019, JCPyV—human polyomavirus 2 (John Cunnigham virus), BKPyv—human polyomavirus 1, HHV6—human herpesvirus 6, GBV-C—Hepatitis G virus.

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
