# Peer review of "The Possible Role of Pathogens and Chronic Immune Stimulation in the Development of Diffuse Large B-Cell Lymphoma"

_biomedicines, 2024, doi:10.3390/biomedicines12030648_

Round 1

Reviewer 1 Report

Comments and Suggestions for Authors

This study explores the association between chronic bacterial infections and the development of diffuse large B-cell lymphoma (DLBCL), emphasizing the role of bacteria like Helicobacter pylori, Campylobacter jejuni, Borrelia burgdorferi, Chlamydia psittaci, and others in modulating immune responses and potentially inducing malignancy. It highlights the complex interactions between bacterial pathogens and the host's immune system that may lead to lymphomagenesis, underscoring the need for further research to clarify these relationships and improve treatment strategies. The manuscript also reviews the effectiveness of antibiotics in early stages of infection-related lymphomas and discusses the prognostic implications of bacterial associations with DLBCL. some comments are listed below for authors considerations:

Section 1:

-While the introduction is comprehensive, consider adding more specific details about the distinct clinical characteristics of some DLBCL subtypes, especially those with notable prognostic differences.

section 2

-While the section is generally clear, consider providing brief explanations for acronyms like ITAM, CBM, and NF-kappaB upon their first mention to enhance reader understanding, especially for those less familiar with the field.

section 3+4

-Consider incorporating figures or diagrams to visually represent Toll receptor signaling pathways.

-Consider providing a bit more detail on the JAK/STAT signaling pathway and its role in interacting with cytokines in the microenvironment. This could enhance the reader's understanding of this crucial aspect

-While TIGIT's importance is emphasized, briefly elaborate on how targeting TIGIT in DLBCL clinically proves more beneficial than blocking the PD1 - PD-L1 axis. This would provide additional context for readers less familiar with the specific clinical implications.

-Section 5:

Clarity on Table 2 Associations: For associations marked with +/- in Table 2, consider providing additional context or explanations in the text to clarify the reasoning behind the designation. This can enhance transparency and reader understanding.

Section 6:

Similar to other sections please define abbreviations

Incorporating visual aids, such as figures or diagrams, could enhance the understanding of complex molecular interactions and pathways discussed in the section

- Ensure that all citations in the manuscript are accurate, and the reference list is up-to-date. Additionally, 

General comments

Literature Review:

  • The manuscript lacks a critical and detailed analysis of existing literature. While the information presented is valuable, a more comprehensive integration of recent studies and data could significantly strengthen the arguments.

Example: In the introduction, the mention of recent studies that either support or challenge the associations between bacterial infections and DLBCL is crucial. Including specific references and contrasting findings would provide readers with a deeper understanding of the current state of knowledge.

  • Refine the introduction to better highlight existing gaps in knowledge and the specific contributions of the study.

Example: The introduction briefly touches upon the associations between bacterial infections and DLBCL, but it could be enhanced by explicitly stating the gaps or uncertainties in the current understanding. For instance, specifying areas where the literature remains inconclusive or where conflicting findings exist would set the stage for the manuscript's unique contributions.

Author Response

Dear Reviewer,

Please find attached the response to your questions. A completely revised manuscript is prepared.

Reviewer 2 Report

Comments and Suggestions for Authors

In this manuscript “The Possible Role of Pathogens and Chronic Immune Stimulation in the Development of Diffuse Large B-Cell Lymphoma”, the authors review critical pathogenetic events in the development of this disease and summarize the data available on several pathogenetic viruses and bacteria that may have a potential role in the development of this lymphoma type. The role of B-cell receptor signaling and Toll receptor signaling is discussed, and the causative role of Epstein-Barr virus (EBV), Human herpesvirus-8 (HHV-8), hepatitis C virus (HCV), Human immunodeficiency virus (HIV), and other viruses are explored. Bacterial infections, such as Helicobacter pylori, Campyl- obacter jejuni, Chlamydia psittaci, Borrelia burgdorferi, and other minor pathogens, are also reviewed. The research is meaningful, but there are some problems with your manuscript. The comments and problems are as follows:

1.I think the language of the summary should be revised to make the article more logical and to highlight the research purpose and significance of the review.

2.I think some of the key words in this article need to be reconsidered, the key words can be more concise.

3.In the introduction, it is important to explain what you bring to the table in order to express your originality. The purpose of the study needs to be found in the last paragraph, and more clearly. Please add further information and reasons and modify accordingly.

4.Why the authors summarize The Possible Role of Pathogens and Chronic Immune Stimulation in the Development of Diffuse Large B-Cell Lymphoma.

Comments on the Quality of English Language

the language of the summary should be revised to make the article more logical and to highlight the research purpose and significance of the review.

Author Response

Dear Reviewer,

Thank you for your valuable review and comments. The manuscript has been completely revised. Please find the detailed letter with responses.

Round 2

Reviewer 1 Report

Comments and Suggestions for Authors

The authors are commended for the revisions. No further comments to authors